# The Psychological Burden of Families with Diabetic Children: A Literature Review Focusing on Quality of Life and Stress

**DOI:** 10.3390/children10060937

**Published:** 2023-05-26

**Authors:** Paraskevi Theofilou, Dimitris D. Vlastos

**Affiliations:** 1General Hospital of Thoracic Diseases Sotiria, 115 27 Athens, Greece; 2School of Social Sciences, Hellenic Open University, 263 35 Patra, Greece; 3Lab of Experimental and Applied Psychology, SCG—Scientific College of Greece, 106 73 Athens, Greece; 4Department of Psychology, SCG—Scientific College of Greece, 106 73 Athens, Greece

**Keywords:** diabetes mellitus, parents, children, adolescents, quality of life, stress, family burden

## Abstract

Chronic diseases, such as childhood diabetes mellitus (DM), are a complex and continuous struggle as well as a great challenge both for the children who face the disease and for their parents. DM is characterized by the complex management of therapeutic treatments, thus causing physical and psychological complications infamily members. There are many families who, upon hearing the diagnosis of their child with DM, stand still in front of these new facts as their lives change. All these unprecedented conditions cause parents intense stress and discomfort, leading them to a mental burden, as the only thing that concerns them upon diagnosis is how the family will survive in the face of the current conditions they are experiencing as well as the future of the sick child. The purpose of this brief literature review is to present the research findings related to the psychological burden of families withchildren with DM, focusing on the quality of life and stress.

## 1. Introduction

Diabetes mellitus (DM) is a metabolic disease and a global crisis. It is the most frequent endocrine disease in children (2/1000) and adolescents (5/1000), with children—almost always of the insulinopenic type (type 1)—constituting 5% of the total population with diabetes. Type 1 diabetes is one of the most common chronic diseases in childhood, and its prevalence is increasing worldwide [1].

Type 1 DM is caused by insulin deficiency and must be treated with daily insulin injections. In the general population, it accounts for 10% of all cases but is more common than type 2 DM in children [2].

The proposed treatment recommended for the management and treatment of DM, either for type 1 or type 2, is complex and demanding since it requires daily control of blood glucose levels (at least four times a day), control and regulation of carbohydrate intake, frequent administration of insulin (three to four injections a day or infusion from a pump), changing insulin doses to match eating and activity patterns, and testing urine for ketones when necessary [3,4].

DM affects the lives of the family, children, minors, and adults mentally and physically. Next to the sick child stands the family as well, as the child’s social environmentis affected by the disease, and parents or caregivers must provide daily care to their child. Both the child and their family during the announcement of the disease go through the stages of mourning (i.e., “shock”, “anger”, “retreat”, and “bargaining” [5]), and there are four factors that can disturb the gradual and smooth development of the child with chronic diseases, whether this development is mental, social, or emotional. Each phase of a chronic illness can present children and their families with significant challenges and stressors related to (1) the separation (removal of the child from theirfamily environment), (2) the deprivation ofmobility and diet, (3) the effects of medicaltreatment (nursing care), and (4) the effects experienced by the child from the pain, physical disability, and potential stigma.

The purpose of this brief literature review is to present the research findings related to the quality of life among parents of children and adolescents with diabetes by focusing on the psychosocial well-being entire system.

## 2. Methods

This literature review examined the following research question: How does chronic disease impact the quality of life of families with children or adolescents who have been diagnosed with diabetes? The review was limited to families with diabeticchildren or adolescents. The following keywords and phrases were searched in the Web of Science (WOS), PubMed, CINAHL, Psych Info, and Google Scholar databases: diabetes *, diabetes mellitus *, quality of life* OR health-related quality of life *, child * OR adolescent * OR paediatric*, physical disability *, and chronic disease OR illness *. Articles were included if they have been published between 1995 and 2022 and included a sample of children and adolescents with diabetes. Titles and abstracts were screened, and articles that met theinitial screening criteria were organized in Mendeley. The appropriateness of the included articles was confirmed by an external colleague. After the final selection, references from the included articles were reviewed and additional appropriate studies that were missed through the initial search were added.

## 3. Results

After searching the literature, 682 articles were found. At the end of the first screening, we selected 76 articles eligible for reading the whole text following the selection criteria described above. Finally, after checking the reference lists, 31 eligible articles were selected for the final analysis. A PRISMA flowchart of the selection and screening method is provided (Figure 1).

### 3.1. Quality of Life in Parents of Children and Adolescents with Diabetes

The diagnosis of a chronic illness or disability such as diabetes in a child leads to family upheaval and disorganization. The effects of this stress on parents can not only negatively affect the mental health of the parents but also affect the health of the child with diabetes. There have been several quantitative and qualitative studies in Greece and abroad thatstudied the burden and mental health of parents of children with diabetes.

Several studies have been carried out to measure the quality of life of children with diabetes and the impact of the disease on their families. One study took place in Saudi Arabia, in which 315 adolescent patients, aged 12–18 years, and their caregivers participated. The results of the study showed that early adolescents (i.e., 13–15 years old) had a better quality of life compared to the group of adolescents aged 16–18 years old. This outcome has been attributed to the fact that adolescents feel more autonomous and handle the insulin regimen by themselves with less parental influence, resulting in poor self-care management. Conversely, the younger age groups showed a better quality of life. Throughout this study, the concern of parents of children with diabetes mellitus was highlighted in terms of the impact of the disease on the family, as well as regarding the complications and treatments related to diabetes. Additionally, the results indicated a negative impact on parents’ psychological well-being [6,7].

Another study carried out in the pediatric hospitals of Patras, Athens, and Ioannina in 2016 aimed to investigate the socio-demographic factors that influence psychopathology, external shame, and the potential experience of shame of 285 parents with diabeticchildren. The findings of this study showed that the duration of the illness is associated with a reduced intensity of the psychological impact on parents over time, as growing children often become responsible for managing the treatment and, thus, partially relieve their parents. Additionally, over time, parents themselves accept the disease as part of their lives, thus reducing the consequences of the psychosomatic shock they suffered after the initial diagnosis, as well as the consequences of internal and external shame. Additionally, the gender of the child play an important role, as girls show lower rates of self-esteem and areless emotionally stable compared to boys. The same applies to parents, as their well-being is influenced by their place of residence and their educational level [8].

Parentsof children attending secondary schoolshada better understanding of the illness severity. Respondents residing in small towns and who were parents of girls presentedhigher scores ofpsychopathologycompared to the parents of boys. An equally important finding of this study wasthat the origin of the child’s name from the respondent parent’s grandparent wasan influencing factor of the scale of shame (ESS), which had a greater influence onthe residents of small towns. In conclusion, through this research, it was pointed out that supportive interventions for parents of children with diabetes should be considered as factors influencing the place of residence, the educational level, the gender of the child, and the origin of the child’s name [8].

Another study was conducted in a pediatric endocrinology unit in Turkey and involved a total of 64 children aged 8–12 years and 85 adolescents aged 13–18 years with type 1 diabetes. The PedsQL was used to measure the quality of life of the children, adolescents, and their parents. The parent form had questions about marital status, education, and family income. There were also questions about whether the children had hypoglycemic attacks and how many, as well as how many times they needed to be taken to the hospital because of diabetes, including the days of hospitalization required. The results forparents and childrenwere relatively the same. The school life scale and the emotional scale were low, whereasthe social scale was high, indicating the social needs of these children and their parents and that such needs must be met. Additionally, this finding may be related to the fact that families have to engage in the school system whilst seeking proper care. Therefore, school safety is a crucial issue. A correlation was also observed between the Child and Adolescent Quality of Life Scalescores. A negative correlation was found between the number of children and the quality of life of children and parents [9].

A study carried out in Sweden with the participation of 130 families from four diabetes centers also examinedthe quality of life of parents and children (5–18 years) with diabetes. The results of the research showed that parents were more likely to identifywith diabetes than their children and participated to a large extent in the management of diabetes. In fact, parents shared responsibility for the management of the disease, and this is something that burdened them to a great extent. Parents were the ones who perceived that the disease limits their children from many activities. Additionally, during the research, it was observed that there was a difference in the perception of the psychological influence ofthe child’s gender. In particular, girls often “freeze” when faced with psychological difficulties to a greater extent than boys [10].

Other researchers have studied the effects of parents’ overprotection towards their children and the problems that arise. The Parent Protection Scale was used to measure parental overprotection. A total of 43 children (15 boys and 28 girls) aged 8–12 years and their mothers (28–48 years) participated in the study. The results showed that parental stress was associated with high levels of depressive symptoms among children [11].

Another study sought to assess the quality of life and health of young children with type 1 diabetes compared to their healthy peers, considering family functioning and mothers’ duties. A total of 113 mothers provided data for the study, 28 of whom had preschool children with type 1 diabetes. There were no differences in the quality of life and family functioning between children with diabetes and healthy children. Mothers of children with diabetes mellitus reported lower levels of resilience and depressive symptoms to a greater extent than mothers of healthy children. Regression analysis revealed that the mothers of children with diabetes mellitus were affected by their child’s disease, resulting in depressive symptoms, and the overall functioning of the family was largely dependent on the child’s illness. Mothers experience great discomfort with diabetes management; however, the family can function properly and young children can live similarly to their healthy peers. In conclusion, based on this study, the quality of life of preschool children is affected by the depression of their mothers. However, as mentioned above, there was no difference in functioning between families that have a child with diabetes or not, as this percentage is related to the child’s quality of life, family functioning, mother’s resilience, life satisfaction, and well-being [12].

### 3.2. Family Burden

Diagnosing diabetes in a child often causes reactions of intense anxiety, shock, sadness, grief, frustration, guilt, and depression in parents. Several studies have been conducted to study the burden onparents of children with diabetes as well as to understand why thesefeelingsare more intense in these families. A study carried out in Norway aimed to identify the similarities and differences between mothers and fathers regarding the degree of emotional distress that arises from caring for a child with diabetes. A total of 103 mothers and 97 fathers participated, as well as 115 children aged 1–15 years. The results showed that both mothers and fathers reported that emotional burden was related to long-term anxiety about their child’s health. Mothers reported a high burden related to the medical care required for their children, feeling more emotional stress than fathers did. Nighttime scores are significantly related to parental burden and emotional stress experienced by both genders [13].

Another study conducted by Hatton et al. [14] identified three different phases of parental coping with an infant or young child diagnosed with diabetes. In the long-term phase, increased stress, sadness, frustration, anger, and loss of control have beenreported by parents. These feelings become more intense as the child grows and has other deceptions about the consumption of food, which leads to confrontations between the parents and the child.As a result, parents often want to entrust the care of their child to others, in case of increased blood glucose levels and the fear of hypoglycemic shock, but also fear about the future of their child.

Faulkner [15] studied parents and siblings of children with diabetes and found that mothers of children with diabetes remembered vividly the time of initial diagnosis and reported the intense emotions they experienced, such as shock, anger, and denial. Many mothers mentioned the pain they felt every time they measured their children’s glucose levels and how painful they found this process.

Dashiff [16] in her study found the great burden of mothers who, as they reported, are overwhelmed by grief. Grief apparently exists in these families, but it is unclear whether it is caused by the turmoil brought on by adolescence or if it is a part of the ongoing turmoil brought on by the disease. Eakes et al. [17] found adisparity experienced by parents of children with diabetes as they face daily situations that remind them that their children have special needs and cannot have a completely normal life. Parents of children with diabetes often describe the situation as a period of constant confusion, guilt, fear, and in some cases intense sadness [18].

Another study, conducted in 2007involving 17 parents of children with diabetes amongwhom 10–17 years had passed since the diagnosis of the disease, found that parents adapted to the needs of diabetes management, but most did notemotionally accept the fact that their child was diagnosed with diabetes. They experienced many times a resurgence of their grief during the development of their child, and many of the mothers were upset during the interviews, even though 10–17 years had passed since the initial diagnosis. Mothers processed their emotions more than fathers did, and both parents experienced feelings of sadness, pain, and anger [19].

Another study approved by the Yale School of Nursing Human Research Review Committee and involving 28 mothers of children with diabetes addressed the burden and emotions experienced by families of children with diabetes. The results of this study showed that mothers of children with diabetes are constantlyvigilant, have constant concerns about hypoglycemia, and care intensively for their children. There were many mothers who felt insufficient support for this, and the necessity of supportive healthcare policies wasemphasized inthe study [20]. In the same manner, a study thataimed to examine the daily experiences of mothers of children with diabetes showed their concern regarding the fear they experience in case of a hypoglycemic shock and a seizure episode [21]. Another study involving 27 participants addressed the burden experienced by the parents of children with diabetes. The conceptual framework of this study was based on insights derived from Orem’s self-care deficit theory. The results showed that parents recognized that their family life changed when they learned about their child’s illness, and they feltconstant confusion about the course of the illness, as well as about the future of their children [15].

Another qualitative study interviewed 30 parents about their experiences of raising a child with diabetes. This analysis revealed sixthemes that focused on parental concerns as follows: (1) long-term complications of diabetes, (2) challenges from disease management, (3) financial concerns, (4) child mental health, (5) health coverage, and (6) experiences with people outside the family and at school [22]. Parents of children with diabetes struggle to balance diabetes management and standard parenting, and these challenges can lead to parental burnout. Although burnout is a familiar term, in the diabetes literature there is a dearth of references to the exhaustion experienced by the parents of these children. Such issuesrelated to this exhaustion have been attributed to stress, inability to control diabetes, sadness, and leadinga normal life [23].

Finally, another study by Jean and colleagues aimed to show the way parents communicate with their children suffering from diabetes, as well asexplore the communication patterns that may lead to a better understanding within the family and more efficient problem-solving strategies. The results of the study showed that there wasa large degree of frustration when parents communicated with their children. Both parents and children expressed insecurity and fear in expressing their feelings, which led parents to psychological pressure and permanent stress, which worethem down psychologically [24].

### 3.3. The Stress of Parents

The effects of parental stress often lead parents to develop mental disorders, which also affect the health of the child with diabetes. So, not only does the family affect diabetes but diabetes also affects the family. A longitudinal study conducted with the participation of 132 teenagers with diabetes and their parents showed that the stress of the parents, which is related to diabetes, leads to poor mental health amongthem and their children.In particular, this study examined parental stress and its influence on parents’ mental healthas well as the relationship between parental stress and health-related quality of lifeinchildren with diabetes [25]. Another study of 69 fathers of children with chronic diseases, such as diabetes, studied whether persistent exposure to stress can lead fathers to experience post-traumatic stress. The results of this study showed that fathers with high levels of stress are at greater risk of post-traumatic stress. Finally, the severity of stress was related to the duration of the disease diagnosis. Fathers six months after the initial diagnosis of the disease had more stress than those who had passed a year since the diagnosis [26].

A study investigating the pediatric stress experienced by fathers of a child with diabetes showed that children experience low levels of pediatric stress, but the correlates of paternalstress may have effects on general family functioning as well as children’s behavior. Forty-three fathers of children (aged 2–6 years) with diabetes completed aself-report questionnaire [27]. Another study examined the stress faced by parents and involved 134 parents of children with diabetes. The analysis showed that the stress experienced by parents is multifaceted and related to the age of the child and the socio-economic status of the family [28]. Finally, another study in which 102 parents participated investigated parental stress and found that parents of children with newly diagnosed diabetes are at risk of developing depression as well as feeling that their lives have changed, resulting in them experiencing constant emotional discomfort [29].

A study by Lowes and colleagues, which aimed to gain a new theoretical understanding of parental reactions and adjustment processes to the disease, showed that the diagnosis of childhood diabetes leads parents to a psychosocial transition. This transition is the grieving process and the stages of adjustment that parents experience. In all these stages, parents experience strong emotions such as stress, depression, guilt, and fear, as they are overwhelmed by uncertainty regarding the outcome of the disease and the life of their children [30].

A descriptive study conducted in 2016 involving 113 parents of children and adolescents with diabetes in Shiraz, using the Depression Anxiety Scales (DASS-21) and the Connor-Davidson Resilience Scale, aimed to identify psychological predictors of resilience in parents of children and adolescents who are insulin-dependent. The results showed that parents had high levels of stress, with 71.4% reporting mild to extremely severe symptoms of depression and anxiety. Additionally, 49% of the changes in resilience weredue to factors such as stress, depression, and life satisfaction. Since nearly half of the parents experienced anxiety, depression, and stress, and there was a correlation between resilience and these psychological variables, parents’ psychological problems, especially depression, may be reduced by improving their resilience [31].

The aim of anotherstudy by Antonio Zayas and his colleagues was to study the levels of depression in parents and to analyze the resilience of these parents. Thirty-oneparents of children with diabetes were evaluated by the JEREZ diabetic association. The results of the study showed that these parents showed moderate levels of depression. It was also found that those who were more resilient had lower levels of depression [32]. Another study by Luo and colleagues was conducted to determine whether resilience moderates the detrimental effects of caregiver burden on the quality of life between caregivers and children with diabetes. The study involved 227 parents, and the results showed that caregiver burden was associated with parents’ quality of life, while resilience was found to be positively associated with the quality of life. Resilience served as a moderator between caregiver burden and mental health. When parents faced a high caregiver burden, the benefit of high resilience to better mental health was evident [33].

Another study referred to by Koegelenberg investigates the properties of resilience and concludes that family communication, the routine of everyday life, and the time a family spends together are important characteristics of resilience and influence the adjustment of families with children diagnosed with diabetes [34]. Ozlem Kara and colleagues studied the relationship between the levels of anxiety and depression of children with type 1 diabetes and the resilience of their parents’ coping attitudes. The study involved 71 parents and children with diabetes, and the results showed that parents with children who have a chronic disease show different coping attitudes. These behaviors cause various effects on children. In conclusion, families that use appropriate coping strategies have a positive impact on their children [35].

## 4. Discussion

Diabetes mellitus is a challenge for patients andtheir families who are suddenly challenged to adapt to a condition that causes changes in the functioning of the whole family, especially when it comes to children or teenagers. Primary health care must provide support to these families so that they can overcome in the smoothest possible way all the transitional stages from the information of the disease to the acceptance of their children’s disease. Parents, as we have seen through the above studies in which they participated, felt strong negative feelings of stress, anger, and pain, which, through appropriate support from experts, can bealleviated in order tolead them to a more functional behavior so that they can accept the disease. Parents who show increased levels of depression, overprotection, and stress increase the likelihood of their children experiencingsuch negative symptoms.

The recognition of the influence of socio-demographic factors on the psychopathology of parents highlights the need for targeted interventions to helpthem become more effective in supporting their children in dealing with diabetes mellitus.

The above observations raise new questions about the psychological processes associated with the occurrence of diabetes mellitus, a disease that inevitably affects the unconscious perceptions and associations of the parent in relation to the child and creates new questions that, to the extent that they cannot answer, create guilt and increase stress and indicators of psychopathology. The sudden adverse changescanproduce feelings of inferiority, which in turn are related to shame, anxiety, and depression [36].

In the context of therapeutic interventions, the role of caregivers is very important because of the care they offer. It is critical to understand their needs and assess the burden on their lives. Thus, appropriate interventions should be implemented to at least limit the burden on caregivers by focusing on the individual factors that contribute to this phenomenon. In this context, it is proposed that research efforts should be made in this direction.

Finally, it must be noted that during the past 30 years, there has been a dramatic change in diabetes managementregarding insulin administration, glucose measurement, and types of insulin. These changes have improved the lives of children with diabetes as well as their families [37]. Moreover, collaborative management among children with diabetes, their family members, and the school faculty must be addressed. In this way, successful experiences for children with diabetes will take place every day.

## Figures and Tables

**Figure 1 children-10-00937-f001:**
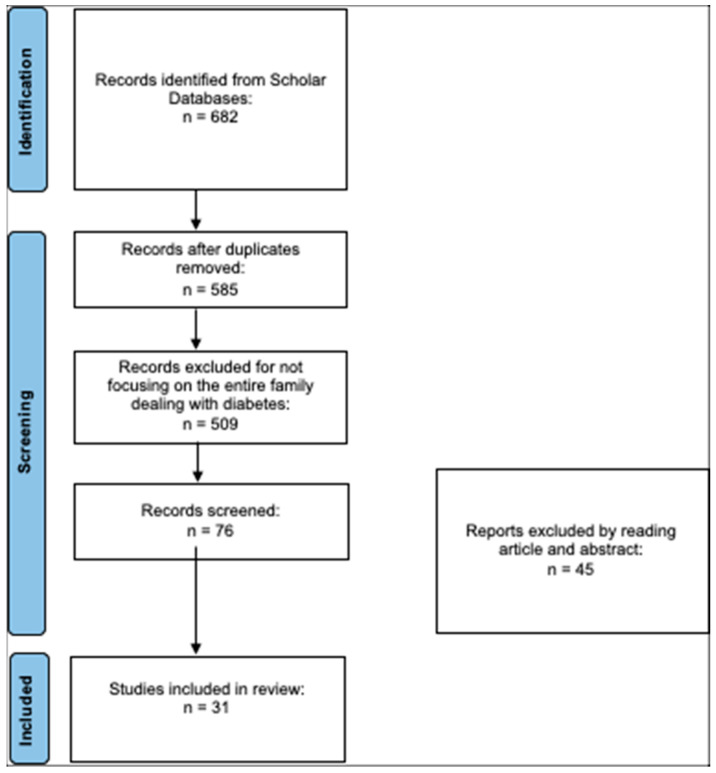
PRISMA flowchart of the literature review.

## Data Availability

Not applicable.

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
