# Peer review of "The Psychological Burden of Families with Diabetic Children: A Literature Review Focusing on Quality of Life and Stress"

_children, 2023, doi:10.3390/children10060937_

Round 1
Reviewer 1 Report
Thank you for the opportunity to review this manuscript about quality of life among parents of children and adolescents with diabetes. Generally, the topic is highly important and literature reviews are needed to shed light on the quality of life of parents of children with diabetes.
However, I have some serious concerns about the lack of a methods section, lack of analysis (or quality assessment) and the too short discussion section. The serious nature of these concerns imply that I cannot recommend this manuscript for publication in its current state. I encourage the authors to keep on working with this important topic. Below I add some comments for the authors to consider, should they decide to revise the manuscript.
Line 21-40 is mainly background information on adults -and on type 2 diabetes. This knowledge is not that relevant to a population of parents of children with (primarily) type 1 diabetes. I recommend deleting this part.
The term ‘diabetic’ should, be changed to ‘persons with diabetes’ in line with the current focus on ‘language matters’ (http://oro.open.ac.uk/55720/1/Cooper_Lloyd_et_al_Language%20Matters%20Position%20Statement-2018-Diabetic_Medicine.pdf)
Line 44: The presented four factors suggested to be disturbing the gradual and smooth development of the child with chronic diseases are not explained further and are based on a publication from 1998. The reason for using this rather old publication should be explained or a more updated publication should be used to illustrate the factors potentially impacting on the development of a child with chronic illness. A more updated approach would probably also include e.g. stigma.
There is no methods section describing how the literature review was conducted. This is a major problem for the credibility of the results.
Section 2 ‘Literature review’: This section is not referenced correctly. Further, the section is purely a description of studies. No quality assessment or metanalysis is conducted.
The discussion section is insufficient.
A native english speaker should review the manuscript.
Author Response
Children-2357573
Response to Reviewer’s 1 Comments (with bold, our responses)
Reviewer 1:
Thank you for the opportunity to review this manuscript about quality of life among parents of children and adolescents with diabetes. Thank you for your valuable feedback and suggestions.
Generally, the topic is highly important and literature reviews are needed to shed light on the quality of life of parents of children with diabetes.However, I have some serious concerns about the lack of a methods section, lack of analysis (or quality assessment) and the too short discussion section. We understand your concerns and we sought to improve our manuscript by adding a distinct and clear methods section in which we explain our search strategies as well as the criteria we implemented for our literature review. Additionally, we extended our discussion.
Line 21-40 is mainly background information on adults -and on type 2 diabetes. This knowledge is not that relevant to a population of parents of children with (primarily) type 1 diabetes. I recommend deleting this part. We appreciate your comment and suggestion. We ‘readjusted’ the introduction by deleting this part and focused on both types of DM which greatly affect the patients as well as their support system and families.
The term ‘diabetic’ should, be changed to ‘persons with diabetes’ in line with the current focus on ‘language matters’(http://oro.open.ac.uk/55720/1/Cooper_Lloyd_et_al_Language%20Matters%20Position%20Statement-2018-Diabetic_Medicine.pdf) We apologize for that. Now we use the correct expression.
Line 44: The presented four factors suggested to be disturbing the gradual and smooth development of the child with chronic diseases are not explained further and are based on a publication from 1998. The reason for using this rather old publication should be explained or a more updated publication should be used to illustrate the factors potentially impacting on the development of a child with chronic illness. A more updated approach would probably also include e.g. stigma. Thank you for the suggestion. We updated this part with an additional in-text citation that reflects these factors as well as social factors.
There is no methods section describing how the literature review was conducted. This is a major problem for the credibility of the results. You are more than right. We added the methods section.
Section 2 ‘Literature review’: This section is not referenced correctly. Further, the section is purely a description of studies. No quality assessment or metanalysis is conducted. We renamed this section to the “Results” section for our literature review.
The discussion section is insufficient. Thank you, the discussion section after your review has been extended and enriched according to our findings.
Reviewer 2 Report
Thank you for the possibility to review the manuscript Quality of life among parents of children and adolescents with diabetes: a literature review.
The topic of this manuscript and the special issue is very important, because many children and families need to cope with diabetes in their daily lives, and knowledge is needed about the factors that contribute to difficulties and strengths in coping.
There are, however, several problems in this manuscript that prevent me to recommend its publication in the current form. The main problem is that the authors do not differentiate between type 1 (T1D) and type 2 (T2D) diabetes. T1D and T2D are different disorders with different management and different effects on psychosocial well-being of the children and the parents. In the introduction, the reference is about T2D in adults whereas the studies reviewed in the manuscript are mainly about T1D in children.
The second problem is that some of the referred studies are over 20 years old, and the daily management regimen and the technology of T1D management (e.g. the use of continuous blood glucose monitoring, multiple daily insulin injections and insulin pumps) has been changed enormously since then. The newer studies report less negative effects of T1D to the well-being of the children and their families, although there still are issues affecting the quality of life.
The title refers to the quality of life of the parents. However, the topic of the manuscript is now wider; it tells about the psychosocial well-being of the child with T1D and of the parents. They are related, but a sharp focus in the parents’ well-being would make the text run smoother. The text would also benefit if it presented factors related to the child, family, diabetes and the care that are associated with quality of life of the parents. Maybe, instead of referring the studies one by one, it would be more informative to sum up the findings of different factors affecting the parents’ well-being.
In addition, the studies deal with many overlapping concepts, and it is important to define the concepts (for example, quality of life, psychological well-being, family stress, external and internal shame, overprotection) as well as their relationship with each other.
The introduction should better explain and justify parent’s quality of life as a study question. Now, it focuses on diabetes and the factors related to child’s development.
Resilience is a growing theme in the research of chronic diseases. It is good that the authors have included some of the studies in the manuscript. However, it would be important to discuss their implications and relation with parents’ quality of life.
In most parts, English is readable. There are some expression that I did not understand (e.g. 'Secondary school parents').
The reference list should be edited and checked.
Author Response
Children-2357573
Response to Reviewer’s 2 Comments (with bold, our responses)
Reviewer 2:
Thank you for the possibility to review the manuscript Quality of life among parents of children and adolescents with diabetes: a literature review. Thank you for your valuable feedback and suggestions.
The topic of this manuscript and the special issue is very important, because many children and families need to cope with diabetes in their daily lives, and knowledge is needed about the factors that contribute to difficulties and strengths in coping. Thank you, we appreciate your opinion.
There are, however, several problems in this manuscript that prevent me to recommend its publication in the current form. The main problem is that the authors do not differentiate between type 1 (T1D) and type 2 (T2D) diabetes. T1D and T2D are different disorders with different management and different effects on psychosocial well-being of the children and the parents. In the introduction, the reference is about T2D in adults whereas the studies reviewed in the manuscript are mainly about T1D in children. The second problem is that some of the referred studies are over 20 years old, and the daily management regimen and the technology of T1D management (e.g. the use of continuous blood glucose monitoring, multiple daily insulin injections and insulin pumps) has been changed enormously since then. The newer studies report less negative effects of T1D to the well-being of the children and their families, although there still are issues affecting the quality of life. We understand your concern and we sought to make clear that our literature review focuses on previously published work on the impact of DM on families (i.e., parents and children) without distinguishing the types as the literature suggests common psychosocial consequences and decreased QoL when resources are insufficient. Thus, our selected papers for this review are the only papers reflecting the aim of our literature review.
The title refers to the quality of life of the parents. However, the topic of the manuscript is now wider; it tells about the psychosocial well-being of the child with T1D and of the parents. They are related, but a sharp focus in the parents’ well-being would make the text run smoother. The text would also benefit if it presented factors related to the child, family, diabetes and the care that are associated with quality of life of the parents. Maybe, instead of referring the studies one by one, it would be more informative to sum up the findings of different factors affecting the parents’ well-being. As the focus of our review relies on the psychological burden of families in terms of QoL and Stress we re-adjusted the title: “The Psychological Burden of Families having Children with Diabetes: A Literature Review Focusing on Quality of Life and Stress”
In addition, the studies deal with many overlapping concepts, and it is important to define the concepts (for example, quality of life, psychological well-being, family stress, external and internal shame, overprotection) as well as their relationship with each other. We provide a more coherent discussion section now, that we hope reflects the interrelationship between these constructs.
The introduction should better explain and justify parent’s quality of life as a study question. Now, it focuses on diabetes and the factors related to child’s development. Resilience is a growing theme in the research of chronic diseases. It is good that the authors have included some of the studies in the manuscript. However, it would be important to discuss their implications and relation with parents’ quality of life. You are more than right. We changed several parts in our manuscript in order to enrich the overall quality of our work.
In most parts, English is readable. There are some expression that I did not understand (e.g. 'Secondary school parents'). We rephrased this part to “Parents having children who attend secondary schools”.
Round 2
Reviewer 2 Report
Dear Authors,
Thank you for the possibility to read the revised manuscript. The manuscript has been improved. The changes made in the title and the text make the message more clear. The paragraph Method and the flowchart make the study more justified.
However, there are some things that still need editing. The main issue that still needs considering is the dramatic change in the diabetes management during the past 20 years. I suggest that you discuss its effects on the quality of life of the families with diabetes in discussion and perhaps note that the findings of the earlier studies may not be valid anymore.
Some minor points:
Abstract, the last sentence: "... among parents of children and adolescents with DM." Please edit according to the title.
Line 136: "Respondents residing in small towns had higher psychopathology values than parents of girls compared to parents of boys." The sentence is unclear, please edit.
Line 151: "The school life scale as well as the emotional scale was low, the social scale was high." Please interpret the result.
Line 237: "an epileptic episode": Do you mean "seizure" (in severe hypoglycemia)?
Please use a native English speaking scholar to edit the text. There are still some phrases that are not adequate.
Author Response
Response to Reviewer’s 2 Comments (with bold, our responses)
Thank you for the possibility to read the revised manuscript. The manuscript has been improved. The changes made in the title and the text make the message more clear. The paragraph Method and the flowchart make the study more justified. Thank you for your valuable feedback.
However, there are some things that still need editing. The main issue that still needs considering is the dramatic change in the diabetes management during the past 20 years. I suggest that you discuss its effects on the quality of life of the families with diabetes in discussion and perhaps note that the findings of the earlier studies may not be valid anymore. A paragraph was added regarding this issue in discussion.
Abstract, the last sentence: "... among parents of children and adolescents with DM." Please edit according to the title. We have done this.
Line 136: "Respondents residing in small towns had higher psychopathology values than parents of girls compared to parents of boys." The sentence is unclear, please edit. We have done this.
Line 151: "The school life scale as well as the emotional scale was low, the social scale was high." Please interpret the result. An interpretation has been made.
Line 237: "an epileptic episode": Do you mean "seizure" (in severe hypoglycemia)? We have replaced the word.